# Assessing the Movements, Habitat Use, and Site Fidelity of the Giant Freshwater Whipray (*Urogymnus polylepis*) with Acoustic Telemetry in the Maeklong River, Thailand

Thanida Haetrakul [1,2], Teresa Campbell [3], Chayanis Daochai [2,4,*], Nitiwadee Keschumras [2], Tanatporn Tantiveerakul [2], Zeb Hogan [5,*] and Nantarika Chansue [2]

1 Faculty of Veterinary Science, Chulalongkorn University, Bangkok 10330, Thailand
2 Veterinary Medical Aquatic Animal Research Center of Excellence, Chulalongkorn University, Bangkok 10330, Thailand
3 McGinley and Associates, 6995 Sierra Center Pkwy, Reno, NV 89511, USA
4 Faculty of Veterinary Science, Prince of Songkla University, Songkla 90110, Thailand
5 Department of Biology and Global Water Center, University of Nevada, 1664 N Virginia St, Reno, NV 89557, USA
* Correspondence: chayanis.d@psu.ac.th (C.D.); zhogan@unr.edu (Z.H.)

**Abstract:** We used acoustic telemetry to assess site fidelity, movement patterns, and habitat use within the lower Maeklong River, Thailand, by 22 giant freshwater whipray *Urogymnus polylepis*. This study is the first of its kind for this rare, endangered species, and it begins to fill critical information gaps about its ecology and life history. Study animals were tagged subcutaneously in the dorsal area with acoustic transmitters and tracked for 12 months using a fixed array of eight receivers. Receivers were positioned within an 18 rkm section of the Maeklong, starting approximately 4 rkm upstream from the mouth. We calculated individual residency indices, tracked longitudinal movements, and assessed annual and seasonal patterns of site use. We also investigated spatial use patterns, diel activity patterns, and relationships of temperature and site fidelity. We detected 86% of our tagged whiprays, 53% of which were detected for the majority of the study period. Whiprays exhibited high site fidelity across seasons. Individuals made long longitudinal movements within the site for much of the study period but seemed to remain closer to the estuary during the second half of the rainy season and early winter. All receivers had large numbers of detections, but upstream receivers were visited for longer time durations than downstream receivers. A receiver in the middle of the study area had the highest detection rate, but low detection durations, suggesting that this receiver is in a migration corridor. A mix of immature and mature males and females was present in the site throughout the study period, suggesting that this area is not exclusively a breeding or nursery site. However, the high site fidelity suggests that this is an important aggregation site for the species. Effective conservation measures may include conservation zones and a shrimp reintroduction program upstream, ensuring fish passage through the middle of the site, and regulating traffic and pollution downstream. We found acoustic telemetry to be an effective study method and encourage its use to improve understanding of the giant freshwater whipray.

**Keywords:** giant freshwater stingray; *Himantura chaophraya*; *Himantura polylepis*; Southeast Asia; endangered species conservation; migratory fish; megafauna; tropical freshwater diversity

## 1. Introduction

The giant freshwater whipray *Urogymnus polylepis* is a large-bodied stingray native to waters of eastern India and Southeast Asia [1]. Range wide, the species is listed as endangered by the International Union for Conservation of Nature (IUCN), and the Thailand population is listed as critically endangered [2,3]. It is well known for its large size

(fishers report individuals as large as 600 kg [4]), and has received recent media attention from a record-breaking specimen caught and released in Cambodia in June 2022 that was officially recognized as the world's largest freshwater fish [5]. Despite this, almost nothing is known about its biology, ecology, or life-history traits [1]. Mounting evidence suggests that multiple populations across its range are in a severe state of decline [1]. The whipray faces a variety of threats, including fishing, pollution, habitat destruction, and lack of legal protection [1]. Therefore, research is urgently needed to fill knowledge gaps and inform management strategies and conservation priorities. An urgent priority is to identify critical habitats and population strongholds that should be given immediate protection.

In Thailand, the giant freshwater whipray received protection in 2018 under the Wildlife Preservation and Protection Act. The species occurs in multiple rivers throughout the country, but no information exists on its population density, abundance, habitat use, or movement and residency patterns [1]. The Maeklong River in the southwest corner of central Thailand may hold the most robust population, as evidenced by large sample sizes, news reports of large individuals [1], and regular catches of mature individuals by recreational catch-and-release anglers (N. Chansue, pers. comm.). As such, this river and its population of giant freshwater whipray are deserving of research attention and protection.

To help determine the importance of the Maeklong River to Thailand's giant freshwater whipray population, we studied movement and residency patterns within an 18-river-kilometer (rkm) portion of the lower Maeklong River where it empties into the Bay of Bangkok. Animal movements often reflect ecologically important behaviors, such as food acquisition, predation avoidance, mating, and locating nursery areas [6]. These movements can also be influenced by environmental factors [7]. Acoustic telemetry is recognized as a useful tool for defining home range, habitat use, and distribution [8], but very few studies have been conducted with batoid elasmobranchs (skates and rays) [9–13]. One study of a closely related species, the freshwater whipray *Urogymnus dalyensis*, in Australia found high site fidelity to a small part of the Wenlock River and that whipray movements were influenced by both the diel and lunar cycles [9]. These types of insights are important for defining protected areas and seasonal fishing closures.

The objectives of our research were to (1) describe spatio-temporal trends in the giant freshwater whipray's use of the lower Maeklong River by assessing site fidelity, movement patterns, and habitat use, (2) assess diel patterns of presence events, and (3) examine the relationship between site fidelity and water temperature. We used acoustic telemetry to determine areas of high use in the lower Maeklong River, how long the whiprays remained in this area throughout the year, and whether measured environmental parameters were correlated with residency patterns. This work will help to inform conservation and management practices in the Maeklong and for the broader population of giant freshwater whipray throughout Thailand, as well as Southeast Asia. This research is the first study of its kind for this species, and we hope it will pave the way for more ecological work and important steps toward conservation.

## 2. Materials and Methods

### 2.1. Study Site and Receiver Deployment

The research was conducted in the Maeklong River, Thailand. The Maeklong is located in southwestern central Thailand. It originates at the confluence of the Khwae Noi and Khwae Yai rivers in Kanchanaburi and flows 130 km into the Bay of Bangkok. The receivers were deployed in an 18 rkm stretch of the lower part of the river in Samutsongkram, Thailand (Figure 1). This part of the river has multiple uses, including for recreation, ecotourism, and aquaculture and as a harbor for fishing vessels and public use. Near the estuary, the primary use is as a harbor, while upstream, aquaculture, recreation, and ecotourism become more important (Figure 1; Table 1). Near the estuary, boat traffic is more concentrated, and the riverside areas are more developed with residences, busi-

nesses, factories, and shipyards. Contamination from toxic chemicals and heavy metals may be more prevalent in this area, which could be problematic for the giant freshwater whipray [14]. The upstream portion of our study site has been selected for a shrimp reintroduction program, which may benefit the giant freshwater whipray as shrimp is a key prey item [15].

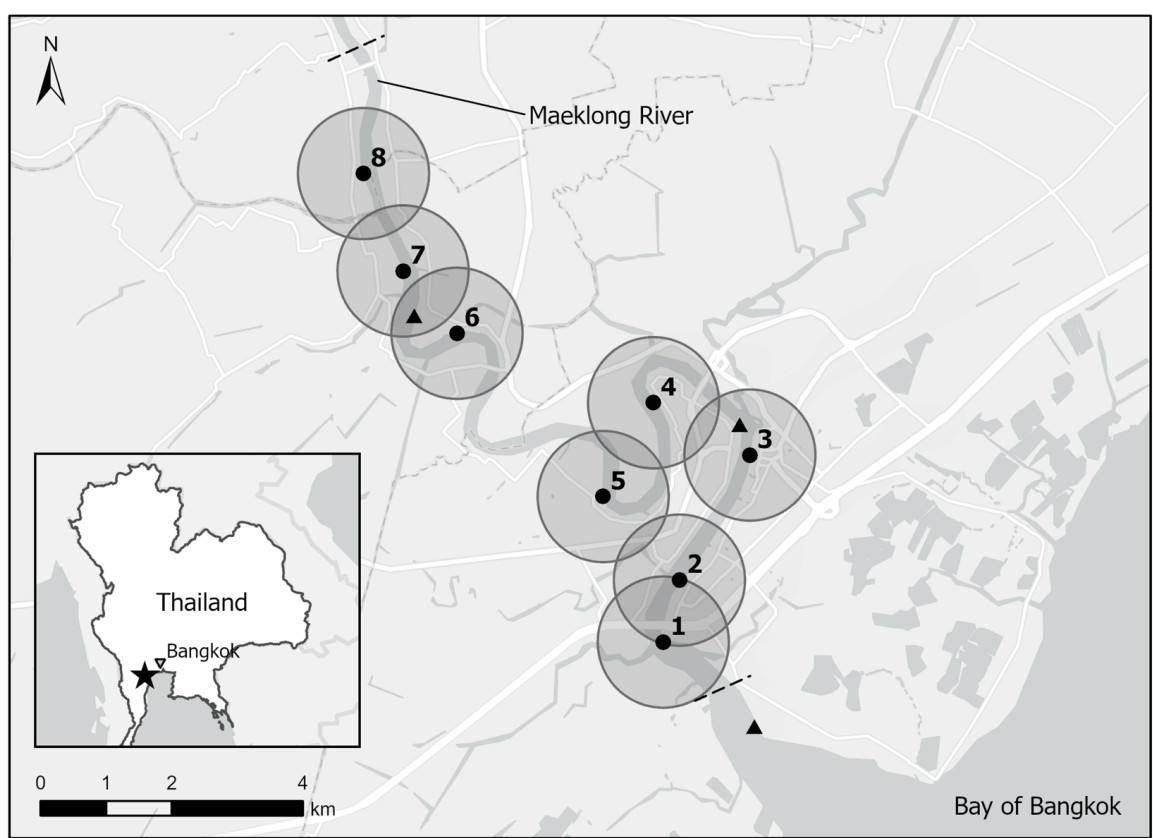

**Figure 1.** Study site showing the acoustic receiver array in the Maeklong River, Thailand. Black dots represent receiver locations. Gray, semi-transparent circles show the lower end (1000 m) of the receiver detection range (1000–2000 m). Dashed lines across the river demarcate the area where study animals were captured. Black triangles show locations of water quality monitoring stations (one additional location is north of the study site out of view). The star in the inset map shows the location of the study site.

**Table 1.** Receiver locations. Primary site use: FH = fishing harbor; PH = public harbor; AQ = aquaculture; RC = recreation; EC = ecotourism. Distance from River Mouth is in rkm.

| Receiver | Lat WGS84 | Long WGS84 | Approximate Distance from River Mouth | Location | Primary Site Use | Deployment Date |
|---|---|---|---|---|---|---|
| 1 | 13.381447 | 99.984996 | 4 | Near estuary | FH | 18 March 18 |
| 2 | 13.389970 | 99.987304 | 5.5 | Mussel Pier | FH | 18 March 18 |
| 3 | 13.407168 | 99.997295 | 7 | Marine Department | PH | 18 March 18 |
| 4 | 13.414427 | 99.983757 | 10 | Baan Tai Had Resort | AQ/RC | 18 March 18 |
| 5 | 13.401533 | 99.976614 | 12.5 | Wat Tai Had | AQ/RC | 18 March 18 |
| 6 | 13.424104 | 99.956114 | 18 | Chaosamran Restaurant | RC/EC | 18 March 18 |
| 7 | 13.432719 | 99.948545 | 20 | Wat Bang Khae Noi | RC | 3 April 18 |
| 8 | 13.446162 | 99.943028 | 22 | Wat Bang Kung | RC | 23 July 18 |

The acoustic receiver array was made up of eight receivers. We used VEMCO VR2W acoustic receivers (VEMCO VR2W-180 kHz, Nova Scotia, Canada). The first receiver was deployed approximately 4 rkm upstream from the river mouth, and the last approximately 22 rkm upstream from the mouth (Figure 1). Each receiver was deployed 1–5 rkm away from each other in strategic locations throughout the river (Figure 1, Table 1). Depending on the type of tag used and environmental conditions, the receivers had a stated detection distance of 1000–2000 m, and so the array was monitoring approximately 20–22 rkm (Figure 1). Although range testing was not conducted for this study, other range testing studies have found that the detection range of these receivers can vary with environmental conditions [16] but also that they can reliably detect tags at 1000 m [17]. If 1000 m is taken as the maximum detection range, then a small gap in the middle of the study area would not be covered (Figure 1). However, we do not expect this to impact the implications of this study. The width of the river in the study area was typically less than 200 m, making the chance of receivers missing tagged whiprays as they passed by very low. River depths have not been measured in the study area, but unpublished data suggest Maeklong River depths range from 5 to 60 m, and fishers in this area describe depths from 5 to 20 m (Chayanis Dochai, pers. comm.).

The receivers were deployed in March 2018 and remained in fixed locations throughout the study period. Receivers were attached to float stations with plastic mesh (Figure 2) and moored one to two meters (m) below the water surface [18]. Float stations were located 5–10 m from the shore, and their positions were adjusted according to water level. Receivers were monitored daily by local residents.

Receivers were deployed for 12 months, from March 2018 to February 2019, and collected data continuously. Data were downloaded every month in the field using a laptop computer with the VUE software (version 2.4.2; VEMCO 2018) and the VEMCO Bluetooth communications kit (VEMCO, Bedford, Nova Scotia, Canada). VUE software is used to communicate with the receivers and download, organize, and process the detection data.

## 2.2. Animal Capture, Work-Up, and Tagging

This work was conducted under Institutional Animal Care and Use Committee (IACUC) protocol 1831064. Between 18 March and 23 July 2018, a total of 22 giant freshwater whiprays were captured within the fishing area (Figure 1) using circle hooks, 200 lb monofilament line, and tuna-style fishing rods. When captured, the whiprays were placed in net-like soft canvas and restrained using a non-anesthetic technique (Figure 2). The sting was wrapped with commercial Coban self-adherent wrap to protect the researchers. Whiprays were then transported to the bank and transferred into an inflatable swimming pool that was 2.5 m wide and 40 cm deep and filled with aerated water (Figure 2). Due to their large size, whiprays were handled in ventral recumbency rather than dorsal recumbency, which is commonly used with elasmobranchs to induce tonic immobility [19], because we felt that it would be less stressful on the animals.

Each whipray was measured for total length (TL), girdle length (GL), disc width (DW), and body weight (WT; Figure 2). Total length was measured from the tip of the snout to the end of the tail; girdle length was measured from the tip of the snout to the pelvic girdle; and disc width was measured between the tips of the wings. Girdle length was measured because of previous observations of whiprays with damage to their tails that would affect the total length measurement.

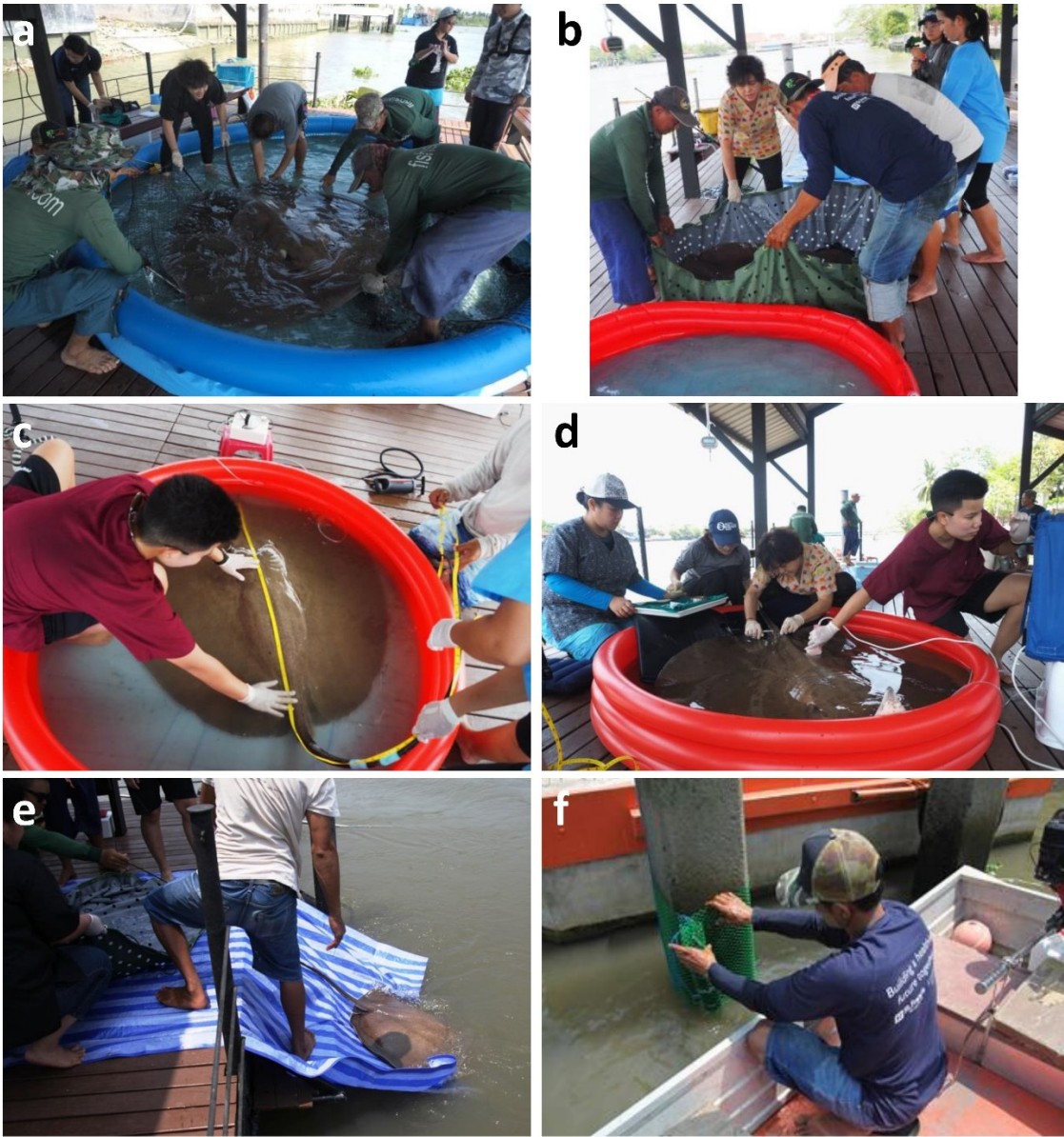

**Figure 2.** Photographs of field procedures, including whipray capture, tagging, and sampling, and acoustic receiver deployment. Panels: (**a**) pool for holding whiprays; (**b**) transportation of a whipray; (**c**) measuring a whipray; (**d**) a whipray being tagged with an acoustic transmitter and scanned with ultrasound; (**e**) release of a whipray back to the river; (**f**) deployment of a receiver.

Sexual maturity was determined by calcified claspers [20] and scanned by My SonoU6 portable ultrasound with LN5-12 electronic linear array transducer or C2-8 curved array transducers (Figure 2). Ovarian or testicular status were assessed for sexual maturity [21,22]. Presence of embryo implantation or fetus was determined for both sides of the uterus [23].

Whiprays were tagged with passive integrated transponder (PIT) tags to provide a secondary form of identification if the whipray was recaptured. The PIT tag was inserted subcutaneously in the left precaudal area [24].

Each whipray was then tagged with an acoustic transmitter that communicated with the receiver array (Figure 2). We used V9-180 kHz transmitters that were 25 mm long, 9 mm in diameter, and 2.1 g in water. They emitted a code in 5–30 s randomized intervals and had an expected battery life of 12 months (VEMCO).

A local anesthetic (lidocaine) was applied to the right distal area, cranial to the pelvic girdle [9]. The whipray's respiratory rate was monitored by observing the movement of the spiracle. An air cushion pad was placed under the abdomen to lift the surgical area above the water surface. The surgical area was sterilized using betadine solution, and a 1.5 cm incision was made approximately 3 cm lateral to the precaudal midline. The transmitter was inserted under the subcutaneous layer. The incision was closed using monofilament absorbable sutures with 2–4 interrupted mattress stiches and secured with a horizontal mattress stitch [25]. The incision was then coated with New-Skin Liquid Bandage. Once the surgery was completed, the whipray was injected intramuscularly with a prophylactic antibiotic.

Total handling time ranged from 10 to 12 min, with surgical time usually lasting less than 7 min. After the surgery and treatment, whiprays were released gently back into the river (Figure 2).

### 2.3. Data Analysis

#### 2.3.1. Annual and Seasonal Site Fidelity

Site fidelity was assessed by calculating residency indices (RI) for each whipray. RIs gave a measure of the proportion of the study duration the whipray was detected by the receiver array. RIs were calculated by dividing the number of days detected by the total number of monitoring days and multiplying by 100 (RI = days detected/days monitored × 100) [26,27]. Monthly residency indices (MRI) and seasonal residency indices (SRI) were also calculated, where MRI = days detected during the month/days monitored during the month × 100, and SRI = days detected during the season/days monitored during the season × 100 [27,28]. Seasons were defined as follows: summer = February through May, rainy = June through October, and winter = November through January.

Differences in RI among life-history groups (immature females, immature males, mature females, and mature males) were assessed using an analysis of variance (ANOVA) test with a significance level of $\alpha = 0.05$. If ANOVA determined there were significant differences among groups, Tukey's post hoc test was used to compare differences between each group. If a whipray was never detected by the receiver array after release, it was removed from the analyses because it could not be determined whether it was not detected because of the individual leaving the array or because of tag loss or mortality.

#### 2.3.2. Spatial Variation in Site Use

The whiprays' spatial use of the study site was assessed by examining individual longitudinal movement patterns and by assessing various receiver metrics. Total presence events, cumulative event duration, and average event duration were used to assess intensity of use at each receiver. A presence event was defined as two or more detections recorded on the same day by the same receiver. The duration of a presence event was calculated as the time difference between the first and last detections of a whipray at a given receiver within a day. Each detection corresponded to a pulse emitted by the transmitter, which occurred every 5 to 30 s, that was detected by the receiver. The number of detections, presence events, and presence duration in hours were calculated using the database management and data analysis functions of the VUE software (VUE Software Manual, Version 2.4.2, VEMCO 2018).

Because the number of presence events at a receiver is affected by the length of receiver deployment, and Receivers 6 and 7 had shorter deployments than the others, we used another metric, the detection index (DI), that was independent of receiver deployment duration and allowed direct comparison among all receivers. The detection index accounted for the time that the receiver was deployed, as well as the number of fish that were

available for detection while the receiver was deployed. The detection index was calculated as the number of fish detection days divided by the number of fish days multiplied by 100 (DI = fish detection days / fish days x 100). Fish detection days is the sum of the number of days each fish was detected by the receiver. Thus, if three individual whiprays were detected by the same receiver in one day, the number of fish detection days would be equal to three. Fish days is the sum of the number of days that each fish was available for detection by the receiver. Thus, if a given receiver was operating at the beginning of the study before fish were tagged, then the number of fish days for that receiver was zero up until the day that fish were tagged. If, on the first day of tagging, four fish were tagged and released, and then the next day, three more fish were tagged and released, the total fish days for that receiver at the end of those two days of tagging would be seven. In this way, the detection index for each receiver was independent of the amount of time the receiver was deployed and the number of fish that were available for detection and, thus, could be compared directly across all receivers.

### 2.3.3. Diel Activity Patterns

To get an understanding of diel activity patterns, the number of presence events during the day and during the night were totaled for all whiprays and by sex and maturity (life-history groups). Presence events are indicative of whipray activity because whiprays must move into the detection range of a receiver in order to initiate a presence event. Day was considered to be 0730 to 1930, and night was 1930 to 0730. The ratio of the number of day events to the number of night events was calculated in order to understand whether individual whiprays tended to have more presence events during the day or at night. This ratio was calculated for the entire group of tagged whiprays, as well as by sex, maturity, and life-history group.

### 2.3.4. Relationship between Temperature and Site Fidelity

Water temperature, salinity, and tide (water depth) data were obtained from the Pollution Control Department database [29]. Data were taken from four monitoring stations positioned throughout the lower Maeklong River (Figure 1). Daily measurements were converted to mean monthly values, which were used to evaluate correlations with RI using Pearson's correlation analysis. The monthly temperature means were compared with mean monthly RI for all whiprays and for female, male, immature, and mature groups. Significance of the correlation was accepted at the $\alpha = 0.05$ level.

## 3. Results

### 3.1. Whipray Morphology and Physiology

The whiprays used for this study included 10 males and 12 females, as well as 10 mature and 12 immature individuals (Table 2). Body weights ranged from 13–105 kg (Table 2). Body size measurements ranged from 1.10–4.14 m TL (mean = 2.65 ± 0.75), 0.63–2.00 m GL (1.11 ± 0.38), and 0.72–2.06 m DW (1.19 ± 0.42) (Table 2). Mature females were significantly larger than mature males ($p < 0.05$) for all measurements except for WT, which was not analyzed statistically because of the small sample size for females (Table S1, Figure S1). Immature males and females were not significantly different in body size (Table S1, Figure S1). Summary statistics of body size grouped by sex and maturity may be found in Table S2. One female (tag number 1062) was pregnant with five pups.

**Table 2.** Physiological and morphological characteristics of tagged whiprays. TL = total length, GL = girdle length, DW = disc width, WT = weight. TL, GL, and DW are reported in meters. WT is reported in kilograms. The shading visually distinguishes males and females.

| Tag No. | Maturity | Sex | TL | GL | DW | WT | Tag Date |
|---|---|---|---|---|---|---|---|
| 1058 | Immature | Female | 2.28 | 0.9 | 0.99 | | 23 July 2018 |
| 1059 | Immature | Female | 2.32 | 0.94 | 1.02 | 26 | 22 May 2018 |
| 1063 | Immature | Female | 1.51 | 0.63 | 0.72 | | 20 March 2018 |
| 1069 | Immature | Female | 2.02 | 0.72 | 0.82 | 13 | 21 May 2018 |
| 1071 | Immature | Female | 1.9 | 0.69 | 0.72 | | 18 June 2018 |
| 1073 | Immature | Female | 2.48 | 0.89 | 0.95 | 22 | 23 July 2018 |
| 1057 | Immature | Male | 1.1 | 0.87 | 0.77 | 15 | 5 April 2018 |
| 1075 | Immature | Male | 2.2 | 0.72 | 0.8 | 14.25 | 3 April 2018 |
| 1078 | Immature | Male | 2.24 | 0.87 | 0.92 | 21.5 | 19 June 2018 |
| 1079 | Immature | Male | 2.36 | 0.92 | 0.98 | 28.1 | 19 June 2018 |
| 1061 | Mature | Female | 2.82 | 1.52 | 1.66 | 105 | 23 July 2018 |
| 1062 | Mature | Female | 3.38 | 1.6 | 1.69 | | 19 March 2018 |
| 1067 | Mature | Female | 4.14 | 2 | 2.06 | | 21 May 2018 |
| 1070 | Mature | Female | 3.92 | 1.75 | 2.03 | | 19 June 2018 |
| 1076 | Mature | Female | 3.95 | 1.79 | 1.93 | | 19 June 2018 |
| 1077 | Mature | Female | 2.9 | 1 | 1.1 | 36.7 | 18 June 2018 |
| 1064 | Mature | Male | 2.92 | 1.26 | 1.31 | 62 | 20 March 2018 |
| 1066 | Mature | Male | 2.65 | 1.1 | 1.21 | 44.3 | 24 April 2018 |
| 1068 | Mature | Male | 2.7 | 0.99 | 1.07 | 33 | 4 April 2018 |
| 1074 | Mature | Male | 3.18 | 1.16 | 1.25 | 50 | 23 April 2018 |
| 1080 | Mature | Male | 2.69 | 1.03 | 1.16 | 40 | 20 March 2018 |
| 1081 | Mature | Male | 2.72 | 1.04 | 1.12 | 36 | 4 April 2018 |

*3.2. Spatio-Temporal Patterns of Site Use*

Eleven of the 22 tagged whiprays were detected by the array throughout the majority of the monitoring period, while the other 11 were detected for approximately one month or less (Table 3, Figure 3). Three of the whiprays were not detected again after tagging. Each of the four life-history groups had two or more individuals with long detection periods and a large number of days detected, total detections, and presence events (Table 3, Figure 3). The largest number of days detected (276), total detections (26,700), and presence events (1084), as well as the longest presence time (1299 h), belonged to two mature males (Table 3).

**Table 3.** Results of acoustic telemetry monitoring of tagged whiprays, including total detections, number of days detected, presence events and duration, and residency indices (RI). Mat = maturity: I = immature, M = mature. Sex: F = female, M = male. Presence duration units are hr:min:sec. The shading visually distinguishes males and females.

| Mat. | Sex | Tag No. | Tag Date | Days Monitored | Days Detected | Total Detections | Presence Events | Presence Duration | RI |
|---|---|---|---|---|---|---|---|---|---|
| I | F | 1058 | 23 July 2018 | 204 | 170 | 5942 | 489 | 335:34:00 | 83 |
| I | F | 1059 | 22 May 2018 | 266 | 122 | 4203 | 279 | 277:05:00 | 46 |
| I | F | 1063 | 20 March 2018 | 329 | 0 | 0 | 0 | 0 | 0 |
| I | F | 1069 | 21 May 2018 | 267 | 50 | 1206 | 107 | 171:04:00 | 19 |
| I | F | 1071 | 18 June 2018 | 239 | 3 | 85 | 6 | 16:50:00 | 1 |
| I | F | 1073 | 23 July 2018 | 204 | 11 | 405 | 32 | 38:07:00 | 5 |

**Table 3.** *Cont.*

| Mat. | Sex | Tag No. | Tag Date | Days Monitored | Days Detected | Total Detections | Presence Events | Presence Duration | RI |
|------|-----|---------|----------|----------------|---------------|------------------|-----------------|-------------------|-----|
| I | M | 1057 | 5 April 2018 | 313 | 9 | 843 | 16 | 41:04:00 | 3 |
| I | M | 1075 | 3 April 2018 | 315 | 1 | 24 | 2 | 1:12:00 | 0.3 |
| I | M | 1078 | 19 June 2018 | 238 | 147 | 7062 | 328 | 453:53:00 | 62 |
| I | M | 1079 | 19 June 2018 | 238 | 37 | 465 | 61 | 35:10:00 | 16 |
| M | F | 1061 | 23 July 2018 | 204 | 26 | 1765 | 98 | 88:30:00 | 14 |
| M | F | 1062 | 19 March 2018 | 330 | 0 | 0 | 0 | 0 | 0 |
| M | F | 1067 | 21 May 2018 | 267 | 10 | 131 | 20 | 10:39:00 | 4 |
| M | F | 1070 | 19 June 2018 | 238 | 162 | 3983 | 400 | 354:44:00 | 61 |
| M | F | 1076 | 19 June 2018 | 238 | 191 | 3486 | 482 | 250:58:00 | 80 |
| M | F | 1077 | 18 June 2018 | 239 | 81 | 805 | 143 | 54:34:00 | 34 |
| M | M | 1064 | 20 March 2018 | 329 | 2 | 29 | 2 | 2:49:00 | 0.6 |
| M | M | 1066 | 24 April 2018 | 294 | 276 | 9587 | 1084 | 604:33:00 | 94 |
| M | M | 1068 | 4 April 2018 | 314 | 261 | 26,700 | 912 | 1299:29:00 | 83 |
| M | M | 1074 | 23 April 2018 | 295 | 3 | 32 | 7 | 3:14:00 | 1 |
| M | M | 1080 | 20 March 2018 | 329 | 0 | 0 | 0 | 0 | 0 |
| M | M | 1081 | 4 April 2018 | 314 | 21 | 184 | 41 | 15:25:00 | 7 |

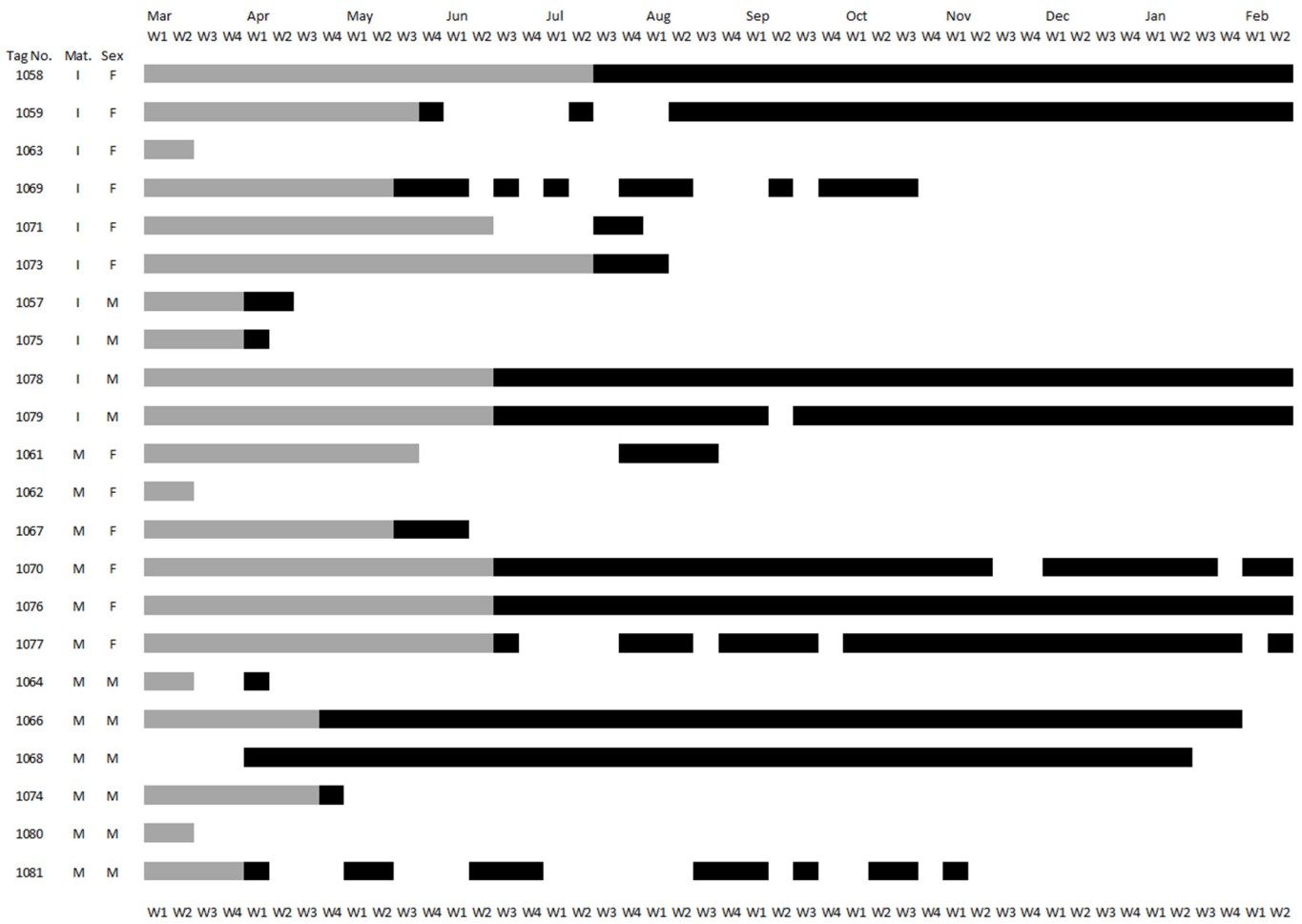

**Figure 3.** Weekly whipray detection data from all receivers during the monitoring period from March 2018 to February 2019. Black rectangles represent weeks during which the whipray was detected by the receiver array. Grey rectangles represent weeks before the whipray was tagged. Mat. = maturity.

Plotting individual whiprays' longitudinal movements showed that many whiprays used the entire study area throughout the course of the monitoring period, with some spatio-temporal variation in use patterns (Figure 4). From approximately August to November, whiprays tended to remain closer to (and may have temporarily moved into) the estuary. Outside of those months, immature individuals tended to remain in the middle to upper reaches of the study area, while mature individuals moved continuously up- and downstream throughout the entire length of the study area (Figure 4).

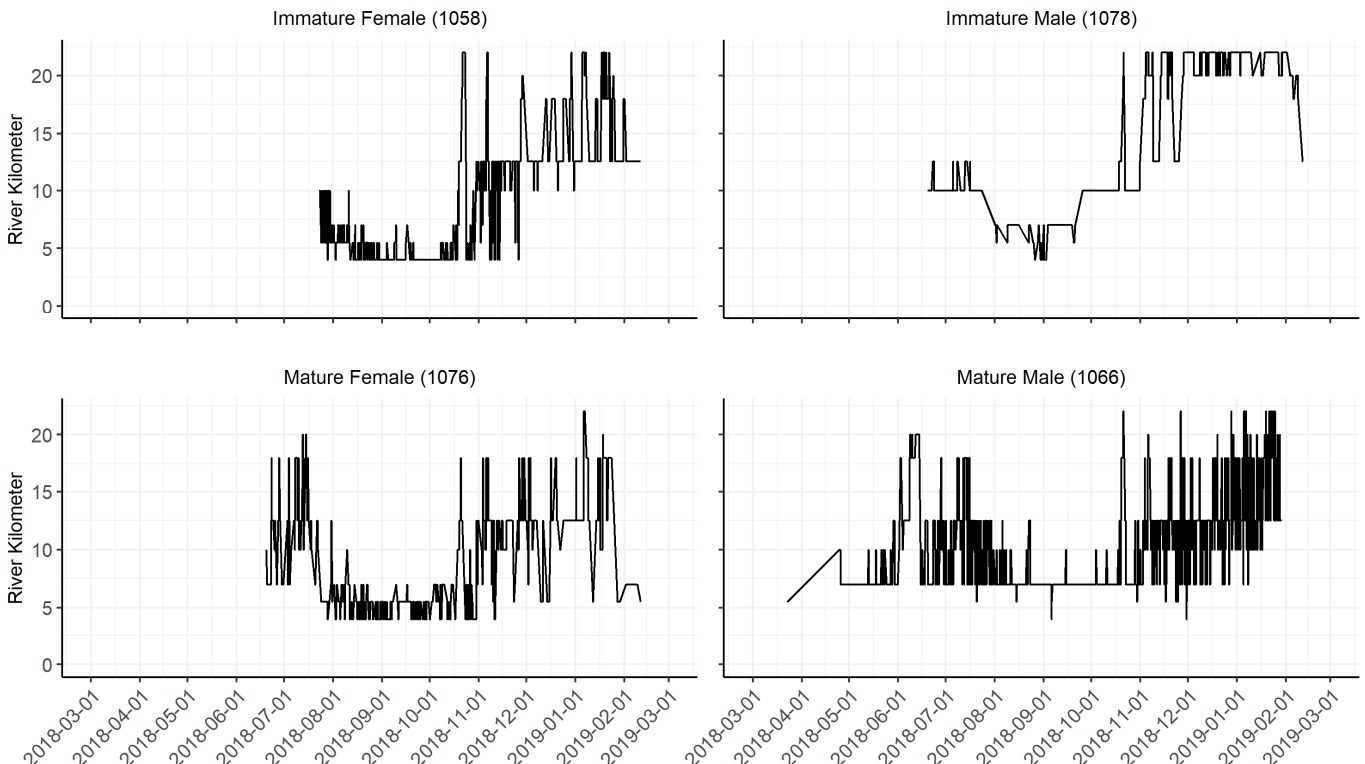

**Figure 4.** Longitudinal movement patterns of tagged whiprays within the study area throughout the duration of the study period. The plots show movements of one whipray (tag number in parentheses) with a long tracking history from each life-history group. Additional whipray movement plots may be viewed in the Supplementary Material. The downstream-most receiver is located at 4 rkm and the upstream-most receiver is located at 22 rkm. Movements outside of this range would not have been detected.

Receivers 6 and 7 had the highest number of total detections (Table S3). Despite being deployed four months after the other receivers, Receiver 8 had more total detections than Receivers 1–5. Receiver 5 had the highest number of presence events, but Receiver 7 had the longest cumulative event duration. The lowest cumulative event durations were from Receivers 1–3 (Table S3).

When accounting for the number of days each receiver was deployed and the number of days each fish was available for detection, it was found that Receiver 5 had the highest detection index and Receiver 8 had the lowest (Table S3, Figure 5). However, average event duration, which is also independent of the length of time receivers were deployed, was highest for Receiver 8, followed by Receivers 7 and 6. Average event duration was lowest for Receivers 3 and 2 (Table S3, Figure 5).

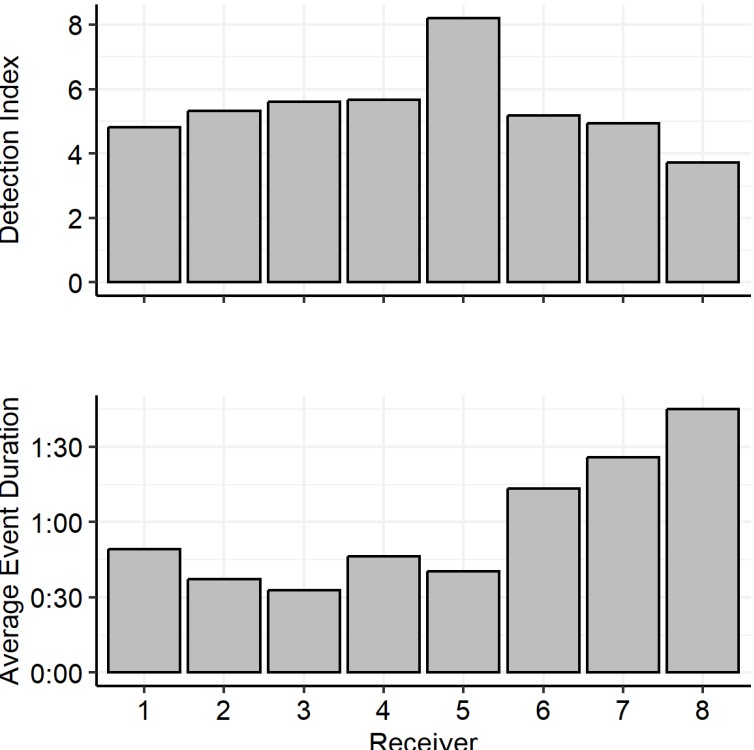

**Figure 5.** Graphs of receiver metrics that are independent of the amount of time receivers were deployed. Detection index = fish detection days / fish days × 100. Average event duration refers to presence events (see methods). Units for average event duration are h:mm.

Clear patterns in receiver use by life-history group were not apparent other than for immature males, which had more presence events on the upstream receivers (Receivers 6–8) than the other receivers (Figure S2). The other life-history groups tended to have more presence events on the middle and lower receivers (Receivers 1–5). Relative to the other groups, mature females had low numbers of presence events on all receivers (Figure S2). Full receiver results by life-history group and by month can be found in Table S4.

There was wide variation in site fidelity (RI) among tagged whiprays (Table 3). The three whiprays that were not detected had RIs equal to zero. Six whiprays had RIs greater than 50, indicating that they were detected by the array for more than half of the days monitored. Overall, RIs ranged from zero to 94, with a mean and standard deviation (SD) of $28 \pm 34$. Removing the three rays with zero detections changed the range of RIs from 0.3 to 94, with a mean and SD equal to $32 \pm 34$. Results of ANOVA determined there were no significant differences in RI among life-history groups (sex: $F = 0.101$, $p = 0.755$; maturity: $F = 0.538$, $p = 0.474$).

Both monthly and seasonal RIs varied among and within months and seasons (Figures S3 and 6). The rainy season had the highest mean and median SRIs, but the difference was not significant according to the results of an ANOVA test ($F = 0.617$, $p = 0.543$). Summer had fewer SRIs greater than 50 when compared to the winter and rainy seasons, but it also had fewer SRIs near zero when compared to winter (Figure 6).

Overall, there tended to be more presence events in the rainy and winter seasons (July–December), especially among females (Figure S4). Males tended to have more presence events in the late rainy and winter seasons. Seasonal patterns of detection were more similar by sex than by maturity, although immature females tended to have more presence events than mature females (Figure S4).

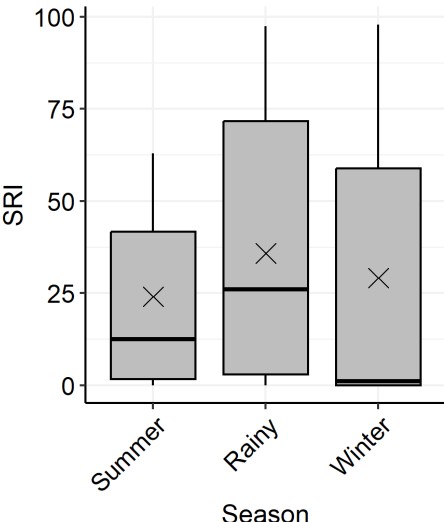

**Figure 6.** Boxplots showing the distribution of seasonal residency index (SRI) data for tagged whiprays. Data are symbolized as in Figure S3. Whiprays that were never detected by the array (n = 3) were removed from the analysis. Summer = February–May; rainy = June–October; winter = November–January. March was not included because three out of four tagged fish were never detected again, and the fourth had an MRI = 0.

*3.3. Diel Activity Patterns*

For all whiprays combined, the total numbers of presence events during the day (2264) and night (2245) were similar (Table S5). The mean ratio of day events to night events for all whiprays was less than one, indicating that individual whiprays tended to have more presence events at night than during the day. However, the mean value of 0.87 was close to one, indicating that the difference in day versus night events was not great. The mean day:night presence event ratios for immature females, immature males, and mature females were all less than one, indicating that these groups tended to have more presence events at night than during the day. The mature males group was the only group that tended to have more presence events during the day than at night (Table S5). Differences were only statistically significant by maturity (F = 4.66, $p = 0.047$), with immature individuals having a mean day:night presence event ratio of 0.70 ± 0.37, and mature individuals having a mean ratio of 1.04 ± 0.30 (Table S5).

*3.4. Environmental Correlates of Site Use*

Water temperature over the study period ranged from 20.3–37 °C with a mean and SD equal to 21.15 ± 2.29. Mean salinity over the study period was 0.32 ± 1.05 g/L. Water tide ranged from 0.32–3.83 m with mean and SD of 2.36 ± 1.11.

Pearson's correlation analyses showed that there was no relationship between mean monthly temperature and mean MRI for all whiprays or for female, male, immature, or mature groups ($p > 0.05$; Table S6). Similarly, no relationship was found between MRI and salinity or tide (Table S6).

## 4. Discussion

*4.1. Site Fidelity*

This is the first research program to directly study the patterns of residency and habitat use of the giant freshwater whipray. Our results have demonstrated that the lower Maeklong River is an important aggregation site for the species in Thailand. The ecological importance of this area was shown by high site fidelity among tagged individuals throughout the year. Several tagged whiprays were detected in the study area year round, leaving for only short time periods. These whiprays included male, female, immature, and mature individuals.

Based on their use of the site, whiprays appeared to belong to three different groups: a high-use group that spent the majority of their time within the site (RI > 50), a mid-use group that spent a shorter amount of time within the site ($16 \leq RI \leq 34$), and a low-use group that spent very little time within the site ($RI \leq 7$; Table 3, Figure 3). Six whiprays belonged to the high-use group (RI = 61–94) and may have been demonstrating home-range behavior [9,30,31]. A seventh whipray had an RI of 46, possibly belonging to this group. High-use whiprays exhibited long periods of presence within the site, from 20 to 37 consecutive weeks, and only short-term absences, indicating that they likely did not travel far from the site (Figure 3). Mid-use whiprays (n = 4) demonstrated longer absences, migrating out of the site for a week to a month before returning. The low-use group was comprised of 11 whiprays, three of which were never detected by the receiver array. These whiprays had very few detections by the receiver array, indicating that they spent most of their time away from the site. It is possible that mid- and low-use whiprays represent either a more mobile portion of this aggregation or members of an adjacent aggregation, possibly migrating through this part of the river to the estuary. Each of these groups had members from all life-history groups, and as such, none were typified by any one sex or maturity stage as in [9].

Sex aggregation and segregation are widespread behaviors among elasmobranchs that result in differential spatial and seasonal habitat use between the sexes [32–34]. Females of some elasmobranch species demonstrate higher site fidelity and aggregation behavior than males in connection with their reproductive behavior and physiology [9,35]. Female freshwater whiprays (*U. dalyensis*) in the Wenlock River, Australia, remained within an 8 km area, while males traveled approximately 60 km downstream to brackish water during the wet season [9]. Other elasmobranch species have also demonstrated segregation of mature and immature individuals [36,37]. The mix of sexes and maturity stages that remained present in our study site throughout all seasons suggests that either this species does not segregate by sex or maturity or that this part of the Maeklong River is not used as a maternal aggregation site or nursery. This area also appears to not be a breeding site because there were no signs of courtship bites on the mature females captured in this study [37,38]. Additionally, breeding aggregations tend to be a seasonal occurrence [39,40]. Therefore, the mixed aggregation observed in this study site must be present for another reason, such as taking advantage of preferred habitat or increased food availability relative to surrounding areas [32,41].

### 4.2. Spatio-Temporal Patterns of Site Use

Whiprays were present in the site in all seasons. There was some evidence to suggest that site fidelity was higher in the rainy and early winter seasons, but differences in MRI and SRI among months and seasons were not statistically different (Figures 6, S3 and S4). Furthermore, the lower summer RIs may have been due to the fact that many whiprays had not yet been tagged in March through May, and the February monitoring period was shortened because of the end of the study period being mid-month. However, it could be that the receding waters in late winter and summer may cause some whiprays to migrate out of the study site and into the estuary or upstream areas, depending on their response to changing flows and salinity gradients [42]. Increased water temperature and decreased dissolved oxygen [43] in summer may also have caused some whiprays to leave the study site for the estuary, although we did not detect a correlation between temperature and MRI. Furthermore, longitudinal movement patterns suggest that whiprays spend more time near the estuary during the second half of the rainy season and into winter (Figure 4). More research is needed to understand the relationship between seasonal environmental changes and whipray movement patterns and broad-scale habitat use.

Whiprays displayed differential use of the site spatially. All receivers had high numbers of detections and detection events, but the upstream area (Receivers 6–8) seemed to hold some importance for whiprays. Although Receivers 7 and 8 had shorter deployments than the others, they received some of the highest numbers of total detections. In general,

the upstream receivers had low to average numbers of detection days and presence events but higher than average total detections, average presence event durations, and cumulative presence event durations, indicating that when whiprays were present in this upstream area, they stayed in the area for longer periods of time than in downstream areas (Table S3, Figure 5). Receiver 5 marked another ecologically notable location. This receiver had the highest total presence events, as well as the highest detection index out of all the receivers (Table S3, Figure 5). This meant that, after accounting for receiver deployment time and the number of fish available for detection, Receiver 5 had the highest detection rate of tagged whiprays. However, its average event duration was low (Figure 5). This may indicate that whiprays regularly pass through this area on their way to and from an upstream aggregation area (Receivers 6–8) and brackish waters in or near the estuary. The longitudinal movement patterns showed many instances of whiprays making long up- and downstream movements within the study site (Figure 4). Downstream receivers (1–4) generally had an average number of detection days and total presence events, but low numbers of total detections and presence event durations (Table S3, Figure 5), indicating that whiprays did not remain as long in the downstream area.

The differential use of the site has different implications for management and conservation strategies. As whiprays demonstrated presence throughout the study site, it is clear that the entire area is worth examining for restoration and management opportunities. However, the upstream area is especially important as shown by whiprays spending more time in this area. Interestingly, the area near Receiver 6 is also the proposed site for a shrimp reintroduction program. As shrimp is a primary prey item for the giant freshwater whipray [15], this program may benefit the population. This area should also be considered for potential fishing conservation zones or gear regulations to help conserve the whipray population [44]. The area around Receiver 5 should be looked at as an important movement corridor, and as such, connectivity between upstream and downstream habitats should be preserved. The downstream area is most impacted by boat traffic, development, and industrial uses, and may be considered for environmental cleanup programs and traffic regulation. The giant freshwater whipray may be sensitive to pollution [14] and boat traffic, which may be why individuals in our study did not spend as much time in this area.

Seasonally and spatially, the site appeared to be used similarly by all life-history groups independent of sex or maturity. This is evidenced by a near-equal sample size for all life-history groups and no statistical differences in RI or seasonal/spatial patterns of site use. A couple of trends were observed that, although not statistically significant, may be ecologically interesting. First, immature females had many more presence events in the rainy season than the other life-history groups, and they also had more detection events than the other groups on Receivers 1 and 2 (Figures S2 and S4). Second, immature males had many more presence events than the other groups on Receivers 6–8 (Figure S2). More research is needed to determine if this is representative of sexual segregation among immature individuals [45].

### 4.3. Life-History Observations

This study also provided much-needed insight into the life-history and ecological traits of the giant freshwater whipray. As in other studies [14], mature females were larger than mature males (Tables S1 and S2, Figure S1). Two individuals over 2 m DW and one pregnant female were captured in our study, showing that the giant freshwater whipray is able to live long enough to grow to large sizes and reproduce in this area. Among immature individuals, there was no statistically significant difference in body size (Table S1, Figure S1). In our study area, we found a near-even sex ratio and ratio of immature to mature individuals. Studies of other elasmobranchs have found even sex ratios [46,47], while others have found female-biased sex ratios [45,46,48]. Both sexes and maturity stages were present year round in our study site, indicating that this combined site use is not isolated to certain seasons. The only indication of possible spatial sexual segregation was among immature males and females (Figure S2).

Individual movement patterns showed that whiprays often made long up- and downstream movements throughout the study site, remaining within smaller areas for certain periods of time (Figure 4). Other elasmobranch research has shown different activity levels in day versus night periods. Nocturnal or crepuscular foraging behavior has been commonly described [49–51]. However, Tilley et al. [31] found greater day-time activity for Atlantic stingrays, which was thought to be due to local environmental factors and predation risk. In our study, we found a similar number of detection events during the day and night for all rays combined (Table S5). Individual rays tended to have more detection events at night, except for mature males, which tended to have more during the day (Table S5). Additionally, mature whiprays were detected more during the day than immature whiprays. However, mean day:night ratios were not greatly different than zero, indicating that, overall, detections were spread relatively evenly across day and night periods. This suggests that whipray movement, potentially for foraging or migration purposes, occurs during both periods. Thus, this species may be described as cathemeral rather than nocturnal or diurnal [52]. As this is a large-bodied species, it may not be as vulnerable to predation and not have as great a need to avoid other species' foraging periods.

### 4.4. Environmental Influences on Site Fidelity

Although environmental factors, such as temperature, salinity, dissolved oxygen, and tide, among others, have been shown to influence elasmobranch movements [7], we did not detect any environmental influences on whipray site fidelity (RI). Temperature has been shown to be an important environmental parameter for elasmobranchs to maintain optimum body temperature [10,53,54], but we found no correlation between RI and mean monthly water temperature. This could be because the range of water temperatures in our study was relatively narrow (20.3–37 °C) compared to the wide thermal tolerance (1–43.4 °C) observed in other batoid species [55]. The Atlantic stingray, *Dasyatis sabina*, showed a preferred feeding and reproductive temperature range of 21.5–31 °C [54]. If the giant freshwater whipray has similar thermal tolerances, then it would not need to take refuge from the temperatures observed in our study. Similarly, we found no correlations between RI and salinity or tide (water depth) as researchers have found for other elasmobranch species [7,9,11,56]. This may be a result of the euryhaline nature of the giant freshwater whipray [57], which may make it less sensitive to these changes. We also may not have detected influences of environmental variables because we used monthly means to compare to monthly mean RIs. Whiprays may be more sensitive to maximum temperatures or daily changes in tide [9] rather than monthly means.

### 4.5. Research Technique

Overall, we found the research technique to be an effective and valuable method for learning more about the ecology and habitat use of this rare, endangered species. The receiver array detected 86% of our tagged whiprays, 53% of which were detected for the majority of the monitoring period (Figure 3). This detection rate is comparable to rates found in other similar telemetry monitoring studies [58–60]. It is also much higher than the recapture rate (5%) found in traditional mark and recapture studies that have been attempted in this area (N. Chansue, pers. comm.). Acoustic telemetry also provides more detailed site use information than mark–recapture studies.

Our high detection rate and long detection periods of tagged fish show that our tagging and processing methods were safe and effective. We note that the subcutaneous dorsal tagging technique used in this study has not been reported elsewhere. Other researchers use ventral peritoneal tagging with the animal in dorsal recumbency to induce tonic immobility [9,10,19,31]. Tonic immobility in prolonged durations can be dangerous to the animal [61,62], and the risk could be greater for large-bodied animals like the giant freshwater whipray. Moreover, insertion of a transmitter into the peritoneal area risks peritoneal infection. The subcutaneous dorsal tagging technique we used was less invasive and did not require an incision to the abdominal cavity. Thus, we felt that it would be

safer and less stressful on the animals. The long detection periods of many of our tagged whiprays support this assumption and the effectiveness of this tagging method.

There is some indication that whiprays may have been negatively affected during handling in March because, out of the four individuals tagged between 19–20 March, three of them were never detected again and the fourth was detected for only two days (Table 3). However, the fate of these whiprays cannot be determined—it could be that they immediately left the receiver array for the estuary or other parts of the Maeklong River. If there was a problem with the handling of the animals, it was quickly resolved because whiprays tagged in April had high numbers of detections (Table 3). We encourage the use of acoustic telemetry to continue learning more about the migratory, reproductive, and aggregation behaviors of the giant freshwater whipray and that researchers learn to safely handle and tag animals from experienced researchers. To date, there have been only a few applications of acoustic telemetry to study endangered fish species in Thailand [58,60,63], and we hope to highlight the benefits of this approach and encourage future research.

## 5. Conclusions

Here, we have clearly demonstrated the importance of the lower Maeklong River to the giant freshwater whipray population and provided valuable insights into the species' ecology and life-history. We hope this research encourages more of its kind so that understanding of this species' ecology and life history can be improved, which can lead to more and better conservation measures. This part of the river may offer a prime location for conservation zones and other fisheries management opportunities that may effectively improve the health of the giant freshwater whipray population. Other opportunities to improve population health include food supplementation through the shrimp reintroduction program upstream and managing pollution and disturbance from tourism and boat traffic in the lower part of the river near the estuary. Although our study was not set up to show the use of the estuary, this habitat is likely important to the species [57], and as such, we believe that conservation measures should be considered for both the estuarine and riverine environments.

**Supplementary Materials:** The following supporting information can be downloaded at https://www.mdpi.com/article/10.3390/w15132311/s1: Table S1: Results of Student's *t*-test analyses of whipray body size as a function of sex and maturity; Table S2: Summary statistics of whipray body size grouped by maturity and sex; Figure S1: Box plots of whipray body size as a function of sex and maturity; Table S3: Summary of receiver detection results; Figure S2: Total presence events by receiver for each life-history group; Figure S3: Boxplots showing the distribution of monthly residency index 9 mri0 data for tagged whiprays; Figure S4: Total presence events by month for each life-history group; Table S4: Receiver presence events by month for each life-history group; Table S5: Number of presence events during the day and night for all life-history groups; Table S6: Results of Pearson's correlation tests for environmental correlates of residency index (RI); Movement Plots of High-Use Whiprays.

**Author Contributions:** Conceptualization, T.H. and N.K.; methodology, N.C., T.H. and N.K.; software, N.K. and Z.H.; validation, N.K., Z.H. and T.C.; formal analysis, N.K., C.D., T.T. and T.C.; investigation, N.C., T.H., N.K., C.D., T.T. and Z.H.; resources, Z.H.; data curation, N.K., C.D. and T.T.; writing—original draft preparation, C.D. and T.C.; writing—review and editing, N.C., T.H. and T.C.; visualization, C.D. and T.C.; supervision, N.C. and T.H.; project administration, N.C.; funding acquisition, N.C. and Z.H. All authors have read and agreed to the published version of the manuscript.

**Funding:** This research was funded through the National Geographic Society's grant program "Science and Exploration in Asia," No. Asia 29-16, and the United States Agency for International Development (USAID) "Wonders of the Mekong" Cooperative Agreement No. AID-OAA-A-16-00057.

**Data Availability Statement:** Data are available upon request.

**Acknowledgments:** We are grateful to the members of Fishsiam who helped us catch the whiprays used in this study, as well as everyone who helped monitor the acoustic receivers. We also thank

Baan Tai Had Resort and Patcharodom Unsuwan, who provided us with the research station site and accommodations.

**Conflicts of Interest:** The authors declare no conflict of interest.

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
