# Peer review of "Assessing the Movements, Habitat Use, and Site Fidelity of the Giant Freshwater Whipray (Urogymnus polylepis) with Acoustic Telemetry in the Maeklong River, Thailand"

_water, doi:10.3390/w15132311_

Round 1

Reviewer 1 Report

General comments:
The manuscript entitled “Assessing the movements, habitat use, and site fidelity of the Giant Freshwater Whipray (Urogymnus polylepis) with acoustic telemetry in the Maeklong River, Thailand” has been reviewed. The paper is well written and well prepared. I have only minor comments to this paper, particularly for the Materials and Methods section. For example, site fidelity was assessed in this study by calculating residency indices (RI) for each whipray. However, it is unclear and needed to specify the used indices detection rate and residency rate. The whiprays were grouped by sex and maturity, but the authors need to show the relation to what they proposed life-history groups. Finally, the most important one is that the number of presence events was related to diel activity patterns. However, the authors need to demonstrate how to use day night ratios to infer the diel activity patterns. Specific comments are provided as below.

Specific comments:

1. Introduction

Line 71-73 We used acoustic telemetry to determine areas of high use in the lower Maeklong River, how long the whiprays remained in this area throughout the year, and whether measured environmental parameters were correlated with residency patterns…

Move this sentence to the objective or the Methods section because this is talking about what will be conducted in the study.

Line 82-83 by assessing site fidelity, movement patterns, and habitat use, (2) assess diel patterns of presence events…

This study assesses site fidelity, movement patterns, habitat use, and diel patterns. The authors may provide additional information regarding to their life history and biological parameters.

2. Materials and Methods

Line 116 -117 Each receiver had a maximum detection distance of 1,000-2,000 m, so the array was monitoring 20-22 rkm of river (Figure 1)…

Provide additional information from literature or previous studies that support a maximum detection distance of 1000-2000 m. As you can see in Figure 1, the detection area may not be fully covered if the range limiting in 1000 m.

Line 184-186

2.3. Data Analysis 2.3.1. Annual and Seasonal Site Fidelity

Site fidelity was assessed by calculating residency indices (RI) for each whipray…

Please clarify the two indices detection rate and residency rate. They should be different to explain the results.

Line 212 calculated using the software package VUE (version 2.4.2)…

Provide detailed information to introduce this software and package VUE.

Line 235 all whiprays and by sex, maturity, and life-history group...

Actually the samples were grouped by sex and maturity. So it would be better to say “by sex and maturity (life-history groups). Explain how the number of presence events relate to diel activity patterns.

Line 235 Overall, there tended to be more presence events in the Rainy and Winter seasons…

So where were they going when not detected by the receivers? Out the detection area or just not detected by the receivers?

4. Discussion

Line 496-499 This could be because the range of water temperatures in our study was relatively narrow (20.3-37 °C) compared to the wide thermal tolerance (1-43.4 °C) observed in other batoid species [53]…

Are you mentioning the wide thermal tolerance (1-43.4 °C) for same species? The area for temperature ranging from 1-43.4°C should have wide coverage in space. The authors may show the known spatial distribution for this species as the demonstration to support this assumption.

Line 362-551 4. Discussion

It would be better to separate the discussion into different parts and highlight the major points of each part.

Reviewer 2 Report

Review comments on Haetrakul et al

 Assessing the Movements, Habitat Use, and Site Fidelity of the Giant Freshwater Stingray (Urogymnus polylepis) with Acoustic Telemetry in the Maeklong River, Thailand

The aim of the study was to use acoustic telemetry to assess the site fidelity, movement patterns and habitat use of the endangered giant freshwater whipray in the Maeklong River in Thailand.  To undertake this project, the authors implanted acoustic transmitters into 22 individuals and tracked their movements for a period of 12 months.  The results of the study showed that the different individuals showed different patterns of movement, but also identified migration corridors and aggregation zones within the estuary.

Overall, the data presented in the paper achieves the study aims and provides an original contribution to improving our knowledge of the movement patterns of the whipray in the Maeklong River.  This data will assist in ongoing management of this endangered species.  The statistical basis for analysis of the movement data of different individuals in the river were appropriate to address the aims of the study. As such, I consider that the paper is suitable for publication. 

There are some minor issues with the references that are detailed below.

 For references 10, 57 and 59, the first letter of the journal title should be capitalized.

 For reference 15, the name of Urogymnus polylepis should be italicised.

 For reference 44, the publication details should be provided.

 For reference 53, the publication year of 2003 is repeated.

Reviewer 3 Report

I think the Giant freshwater whipray study is excellent. Twenty -two rare and endangered fish were tagged and followed in an extensive estuary and river using acoustic telemetry.  The researchers defined habitat use, movement and site fidelity in their effort to conserve this endangered species.  Conservation efforts such as adding shrimp and controlling pollution in relation to tourism are very valuable. The text is clear and well written.  Color photographs and data presentation are very useful and show the fish and researchers working with them. I think this is an excellent paper to include in your journal as a published article.
